# Risk of cutaneous squamous cell carcinoma due to occupational exposure to solar ultraviolet radiation: Protocol for a systematic review and meta-analysis

**Marília Silva Paulo**[1,2], **Cara Symanzik**[3], **Balázs Ádam**[2], **Fabriziomaria Gobba**[4],
**Sanja Kezic**[5], **Henk F. van der Molen**[5], **Cheryl E. Peters**[6,7,8], **Marc Rocholl**[3],
**Thomas Tenkate**[9], **Swen Malte John**[3], **Tom Loney**[10], **Alberto Modenese**[4],
**Marc Wittlich**[11]*

**1** CHRC, NOVA Medical School, Faculdade de Ciências Médicas, NMS, FCM, Universidade Nova de Lisboa, Lisbon, Portugal, **2** Institute of Public Health, College of Medicine & Health Sciences, United Arab Emirates University, Al Ain, United Arab Emirates, **3** Department of Dermatology, Environmental Medicine, and Health Theory, Osnabrück University, Institute for Interdisciplinary Dermatological Prevention and Rehabilitation (iDerm) at Osnabrück University, Osnabrück, Germany, **4** Department of Biomedical, Metabolic and Neural Sciences, University of Modena & Reggio Emilia, Modena, Italy, **5** Amsterdam UMC, Department of Occupational and Public Health, University of Amsterdam, Amsterdam, The Netherlands, **6** BC Centre for Disease Control, Vancouver, Canada, **7** BC Cancer, Vancouver, Canada, **8** University of British Columbia, Vancouver, Canada, **9** School of Occupational and Public Health, Toronto Metropolitan University, Toronto, Ontario, Canada, **10** College of Medicine, Mohammed Bin Rashid University of Medicine and Health Sciences, Dubai, United Arab Emirates, **11** Institute for Occupational Safety and Health of the German Social Accidents Insurance (IFA), Sankt Augustin, Germany

* marc.wittlich@dguv.de

**Data Availability Statement:** No datasets were generated or analysed during the current study. All

## Abstract

Solar ultraviolet radiation (UVR) is the most significant occupational carcinogenic exposure in terms of the number of workers exposed (i.e., outdoor workers). Consequently, solar UVR-induced skin cancers are among the most common forms of occupational malignancies that are potentially expected globally. This systematic review is registered in PROSPERO (CRD42021295221) and aims to assess the risk of cutaneous squamous cell carcinoma (cSCC) associated to occupational solar UVR exposure. Systematic searches will be performed in three electronic literature databases (PubMed/Medline, EMBASE, and Scopus). Further references will be retrieved by a manual search (e.g., in grey literature databases, internet search engines, and organizational websites). We will include cohort studies and case-control studies. Risk of Bias assessment will be conducted separately for case-control and cohort studies. The Grading of Recommendations Assessment, Development, and Evaluation (GRADE) will be used for the certainty of assessment. In case quantitative pooling is not feasible, a narrative synthesis of results will be performed.

relevant data from this study will be made available upon study completion.

**Funding:** The author(s) received no specific funding for this work.

**Competing interests:** The authors have declared that no competing interests exist.

**Abbreviations:** BCC, basal cell carcinoma; DNA, deoxyribonucleic acid; IARC, International Agency for Research on Cancer; KC, keratinocyte carcinoma; MM, malignant melanoma; NMSC, non-melanoma skin cancer; SCC, squamous cell carcinoma; UVR, ultraviolet radiation.

## Introduction

Squamous cell carcinoma (SCC) of the skin belongs to the group of non-melanoma skin cancer (NMSC), also referred to as keratinocyte carcinoma (KC), and is–along with basal cell carcinoma (BCC) and malignant melanoma (MM)–one of the main types of skin cancer. Cutaneous squamous cell carcinoma (cSCC) is caused by deoxyribonucleic acid (DNA) damage that leads to mutations in squamous cells of the epidermis. The rising incidence of skin cancer over the years has made it a significant public health issue: in 2017 there were approximately 7.7 million new cases of KC worldwide [1]. cSCC accounts for approximately 20% of all NMSC cases and poses a lethal threat due to its capacity to metastasize to various organs in the body [2]. The prevention and early identification of cSCC can be aided by understanding individual and environmental risk factors for cSCC and the situations that increase the risk of developing it. Common individual risk factors include fair skin as well as blond or red hair and light-colored eyes, a history of sunburns in childhood as well as in adulthood, a history of precancerous intraepidermal lesions (i.e., actinic keratosis -AK-) or Bowen's disease, a history of previous skin cancer, a weakened immune system (including leukemia or lymphoma patients and patients under immunosuppressants), and some rare genetic disorders (as xeroderma pigmentosum). Dermal exposure to certain chemicals such as arsenic or coal tar as well as exposure to artificial ultraviolet radiation (UVR) (as the use of tanning beds) and to ionizing radiation can cause cSCC. However, probably the most relevant risk factor is exposure to solar UVR, which has been classified as carcinogenic to humans (Group 1) by the International Agency for Research on Cancer (IARC) [3].

Solar UVR is the most pertinent occupational carcinogenic exposure [4–6]. Construction workers, gardeners, fishermen/women, and farmers are examples of jobs with a high frequency of outdoor work and direct occupational solar UVR exposure (i.e., spending the majority of their working hours outside and being thus directly exposed to high levels of UVR); aircraft maintenance engineers, building and construction managers, childcare workers, and police officers are examples of outdoor workers with rather indirect, but potentially significant occupational solar UVR exposure [7, 8]. Epidemiologic data reveal a high prevalence of both BCC and SCC among outdoor workers with several years of cumulative sunlight exposure, demonstrating a strong link between cumulative occupational UVR exposure and the incidence of NMSC [9–12].

Nevertheless, even if a large number of workers spend a significant portion of their worktime exposed to the known carcinogen solar UVR, in many countries' occupational safety and health (OSH) directives and local regulations have not yet acknowledged this work-related health risk [13, 14]. As a consequence, no specific occupational exposure limit values are officially available as a standard [15]. This lack of recognition of the occupational risk obstructs the development of preventive interventions in outdoor workers, whose importance has been demonstrated by a plethora of current studies, even if elevated levels of personal UVR exposure whilst working outdoors is well documented [15–19]. Furthermore, studies have shown that outdoor workers often underestimate the risk of solar UVR exposure [20] and consequently do not adopt sufficient sun-protective habits and behaviors [21]. Recently, it has been reported that the vast majority of professions with time spent outdoors for more than 1 hour are at particular risk [22]. It also was discovered that the unique needs of outdoor workers are rarely considered while developing preventative measures [23]. Different preventative measures are displayed in Table 1.

A meta-analysis conducted in the ambit of a WHO/ILO joint project estimated a significant increase of 60% in the incidence of NMSC for outdoor workers (RR: 1.60; 95% CI: 1.21–2.11),

**Table 1. Measures of primary prevention (i.e., any preventive action aimed at reducing the incidence of cancer in humans), secondary prevention (i.e., preventative action that leads to the detection of precancerous conditions or cancers at an early stage), and tertiary prevention (i.e., actions that take place when the adverse effects are already manifested, including e.g., medical and occupational rehabilitation) [24–27].**

| Stage of the preventive approach | Preventative measures |
| --- | --- |
| Primary prevention | • Preventive efforts and policies taken by governments, employers and institutions, as well as the inclination of certain standards, guidelines, and prevention campaigns;<br>• Establishing a sufficient risk assessment methodology that will be evaluated and updated on a regular basis;<br>• Provision of educational materials (e.g., pamphlets, signs, or phone messages), specific educational training activities, including sun-safety and skin cancer prevention trainings using sunscreen (i.e., broad-spectrum sunscreen with a SPF of at least 30) in uncovered body areas measures according to the so-called TOP principle (technical, organizational, and person-related):<br> • a) technical measures (e.g., sun-shields) organizational measures (e.g. indoor work-breaks or breaks in shaded places and reduction of exposure during the middle hours of the day) person-related measures (e.g., using adequate Personal Protective Equipment [PPE], which comprises a) sunglasses with wide, solar UVR filtering lenses,<br> • b) UVR filtering clothing [i.e., long-sleeved shirts and pants], and<br> • c) headwear [i.e., broad-brimmed helmets or hats with sun shields as well as ear and neck guards]). |
| Secondary prevention | • Occupational health surveillance as periodic medical examinations by trained occupational health professionals and supplementary health controls to be decided on an individual basis (i.e., involvement of other medical specialists, such as dermatologists), screening and early diagnosis. |
| Tertiary prevention | • Medical and occupational rehabilitation aiming at a safe return to work after recovery, including an adequate quality of life, compensations for the occupational diseases diagnosed and properly notified to the authorities. |

without any significant differences for sex and geographical region. The increased risk of cSCC was found in outdoor workers (RR: 2.42; 95% CI: 1.66–3.53) [28], as part of the sensitivity analysis without exploring it per variables. This systematic review will extend our previous work [29] and is aligned with the newest ICD-11 classification where SCC and BCC have, for the first time, different codes and have the possibility to be coded as occupational diseases. Furthermore, we also consider Bowen diseases and AK, not included in previous works. The aim of this study is to evaluate the effect of occupational solar UVR exposure associated with cSCC. To this end, the proposed systematic review will answer the question: What is the relative risk of developing cSCC caused by occupational solar UVR exposure?

## Methods

This protocol has been registered at the International Prospective Register of Systematic Reviews (PROSPERO) on 31 December 2021 under the registration number CRD42021295221. The present protocol was prepared according to the Preferred Reporting Items for Systematic Reviews and Meta-Analysis Protocols (PRISMA-P) [30]. The succeeding systematic review and meta-analysis will be reported in accordance with the Preferred Reporting Items for Systematic Reviews (PRISMA) 2020 statement [31]. If the protocol is modified later on, the date of the amendment will be noted, along with an explanation and reason for the change.

### Eligibility criteria

In the selection of the studies to be included in this systematic review and meta-analysis the Participants, Exposure, Comparator, and Outcome (PECO) criteria [32] outlined below will be considered.

### Participants

The population studied will include adult (above 15 years of age) outdoor workers selected according on the occupation and on criteria of spending more than half of the 8 hours working day outside (outdoors) during daylight hours.

### Exposure

The exposure studied is defined as 1 SED/day, which can be achieved by spending more than 1 hour of the working day outdoors during daylight hours, with occupational exposure to solar UVR. Accepted exposure assessment will be defined as direct and indirect being considered self-reported, from their employment, assessed by a health and safety technician, or the type of dosimeter used. Latitude of residence and ambient UVR levels are excluded.

### Comparator

The comparator group are workers engaged in activities not classifiable as 'outdoor work'.

### Outcome

The primary outcome is cSCC occurrence based on a histologically confirmed medical diagnosis (including registry data). Secondary outcomes are SCC in situ (also referred to as Bowen's disease) and AK based on a histologically confirmed medical diagnosis (including registry data). Subjective self-reporting of the mentioned conditions is not considered adequate.

### Type of studies

The relevant types of studies to be included are cohort studies and case-control studies. Descriptive epidemiological studies, as cross-sectional and ecological studies, case-series and case reports will be excluded as this kind of study design is not adequate to demonstrate causative effects.

### Information sources

Electronic academic databases will be searched for potentially relevant records, including PubMed/Medline, EMBASE, and Scopus. Search terms will follow the above reported PECO criteria outlined, and will include the following factors: workplace, employment, occupational exposure, exposed worker, outdoor workers, workers, occupation, outdoor job, outside occupation, outside work, nonionizing radiation, sunlight, ultraviolet rays, solar radiation, ultraviolet radiation, UV rays, squamous cell carcinoma, skin cancer, and skin malignancies. The search was designed for PubMed and using medical subject headings (MeSH) as well as title and abstract forms (please see Supplementary material). The search will be translated to Scopus and EMBASE following the same structure. We will also search organizational websites (World Health Organization (WHO), International Labour Organization (ILO) and IARC), hand search reference list of previous systematic reviews and selected academic journals. Contacting authors to request data on specific studies will–if necessary–be done by sending an email to the corresponding author and a follow-up email after two weeks. In case that the

corresponding author does not answer after one month, the study will be excluded and document in an appendix file. There will be no date or language restrictions, authors speak Dutch, English, German, Croatian, Hungarian, Italian, Portuguese, and Spanish. Any other eligible papers published in another language will be translated using an online platform, and validated by a colleague from one of the nine academic institutions of the authors. The searches will start upon acceptance for publication of the present protocol and will be re-run just before the final analyses and further studies retrieved for inclusion.

## Study records and selection process

All the studies will be retrieved and imported into a reference manager database where duplicates will be excluded. Afterward, the studies will be uploaded into the software Covidence and pairs of authors will screen the title and abstracts; conflicts will be resolved by a third reviewer where appropriate. Authors will be involved in retrieving the full texts considered and, again, two authors will independently screen the full texts and a third reviewer will resolve conflicts.

## Data items

A data extraction sheet will be developed and piloted until there is convergence and agreement among data extractors. Review authors will independently extract the data on exposure to solar UVR. Conflicts will be discussed and resolved in meetings among the researchers to obtain full convergence. Data on study characteristics (last author, year, country, participants, exposure, and outcome), study design (type of study, measurement of the risk factor and of the outcome, and response rate), study context will be extracted. Information on conflict of interest and potentially confounding factors, (e.g. personal factors, leisure time UVR) will also be extracted.

## Risk of bias in individual studies

For the Risk of Bias (RoB) assessment, we will use the tool described by Romero Starke et al. 2020 [33]. This tool rates the RoB across the recruitment and follow-up, exposure definition and measurement, outcome source and validation, confounding and effect modification, analysis methods, chronology, funding, and conflict of interest. According to this tool, for a study to have an overall low risk of bias, every major domain for risk of bias (recruitment and follow-up, exposure definition and measurement, outcome source and validation, confounding and effect modification, analysis methods) would have to be rated as low risk. For quality assessment, we will use the Grading of Recommendations Assessment, Development and Evaluation (GRADE) approach described by Morgan et al. 2016 [34]. This approach aims to rate the overall certainty of the evidence for the outcomes. The certainty of evidence from observational studies starts at low and assesses the levels of certainty from low to moderate and high [33], while the certainty for etiologic studies, estimating the risk of future events (prognosis) it is also possible to start at high or medium evidence [35]. The GRADE assessment comprises the RoB, inconsistency, indirectness, imprecision, and publication bias across studies.

## Data synthesis

A qualitative narrative synthesis of the aggregated results of the studies included, categorized by type of study, outcomes, outdoor work activity, and world area/region/country will be provided. In the case of sufficiently homogenous studies, quantitative synthesis of the results will be conducted by meta-analytic methods after testing the heterogeneity of the studies. If possible, meta-analyses will pool stratified estimates from cohort and case-control studies using

random-effects model in RevMan 5.3. If more than two risk estimates are available without substantial heterogeneity (i.e. I2 <70%) we will perform secondary analysis by sex (male/female), age, phototype, body site of lesions, occupation (type of occupation as described in the study) country, region (WHO region), latitude (from the city where data was collected), number of years exposed (number of years as outdoor workers), non-occupational exposure, exposure levels (medium or moderate, high), and the adoption of solar UVR protections. Countries will be used as individual countries, e.g., if there is more than one risk estimate per country, we will use both, and we will also categorize countries per WHO world region in a sensitivity analysis. We will extract the estimate with and without adjustment and we will use the most adjusted one to pool our estimates. We expect to find risk estimates adjusting for sex, age and phototype.

## Ethics and dissemination

Because this is a systematic review is based on published studies, no ethical approval or patient permission is required. The systematic review and meta-analysis shall be published in a peer-reviewed international scientific journal (ideally open access). Working group members may also present the findings at national and international conferences.

## Supporting information

**S1 Checklist. PRISMA-P 2015 checklist.**
(PDF)

## Author Contributions

**Conceptualization:** Marília Silva Paulo, Balázs Ádam, Fabriziomaria Gobba, Sanja Kezic, Cheryl E. Peters, Thomas Tenkate, Swen Malte John, Tom Loney, Alberto Modenese, Marc Wittlich.

**Methodology:** Marília Silva Paulo, Swen Malte John, Tom Loney, Alberto Modenese, Marc Wittlich.

**Writing – original draft:** Marília Silva Paulo, Cara Symanzik, Balázs Ádam, Alberto Modenese.

**Writing – review & editing:** Marília Silva Paulo, Cara Symanzik, Balázs Ádam, Fabriziomaria Gobba, Sanja Kezic, Henk F. van der Molen, Cheryl E. Peters, Marc Rocholl, Thomas Tenkate, Swen Malte John, Tom Loney, Alberto Modenese, Marc Wittlich.

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
