## [Decision Letter · Decision Letter 0]

3 Jan 2023

PONE-D-22-18577Risk of cutaneous squamous cell carcinoma due to occupational exposure to solar ultraviolet radiation: protocol for a systematic review and meta-analysisPLOS ONE

Dear Dr. Wittlich,

Thank you for submitting your manuscript to PLOS ONE. After careful consideration, we feel that it has merit but does not fully meet PLOS ONE’s publication criteria as it currently stands. Therefore, we invite you to submit a revised version of the manuscript that addresses the points raised during the review process.

The manuscript has now been reviewed by two outstanding reviewers with strong expertise in the topic. I reviewed the manuscript, too, and I found the protocol interesting and well written. You can find the reviewers comments below, and hereafter my points.
Please add details on the research string to be used in Medline through PubMed, EMBASE, Scopus.Do you plan to contact study authors’ in case you cannot retrieve data from the published reports? If so, please specify it.You declared that you will present the results of the meta-analysis separately for case-control and cohort studies. Being aware of the potential bias of case-control studies, I was wondering why you will not plan to pool together both study types by providing a stratified forest-plot.I cannot understand the sentence “If data estimates from the same study design are available for more than two studies, […]”. What do you mean? For example, the same study reporting separately results by sex, or by age-groups, without providing the overall estimate? Please clarify.Please add more details on the statistical analysis plan; i.e., which model, fixed- or random-effects, which estimator of between-study heterogeneity, any sensitivity analysis, or meta-regression.

We look forward to receiving your revised manuscript.

Kind regards,

Matteo Rota, Ph.D.

Academic Editor

PLOS ONE

Journal Requirements:

Reviewers' comments:

Reviewer's Responses to Questions

**Comments to the Author**

1. Does the manuscript provide a valid rationale for the proposed study, with clearly identified and justified research questions?

Reviewer #1: Partly

Reviewer #2: Yes

2. Is the protocol technically sound and planned in a manner that will lead to a meaningful outcome and allow testing the stated hypotheses?

Reviewer #1: Partly

Reviewer #2: Partly

3. Is the methodology feasible and described in sufficient detail to allow the work to be replicable?

Reviewer #1: Yes

Reviewer #2: Yes

4. Have the authors described where all data underlying the findings will be made available when the study is complete?

Reviewer #1: No

Reviewer #2: Yes

5. Is the manuscript presented in an intelligible fashion and written in standard English?

Reviewer #1: Yes

Reviewer #2: Yes

6. Review Comments to the Author

You may also provide optional suggestions and comments to authors that they might find helpful in planning their study.

Reviewer #1: The protocol that is presented is to conduct a systematic review (and possibly a meta-analysis if there are sufficient quality articles) on the relationship between cSCC and occupational solar UVR exposure. The protocol is very generalized and needs additional detail - bearing in mind the thinking of a protocol is for someone else to follow the 'recipe' to repeat the study in the future.

Please provide more detail on these aspects: by sex, occupation, country or region, latitude, the number of years exposed (number of years as outdoor workers), exposure levels, and the adoption of solar UVR protections. Some detail is given about 1 SED per day, but how will latitude be investigated, for example. Will countries be categorized or analyzed as individual countries, etc. How will occupation be assessed - by category e.g., water and solar UVR exposure, snow and solar UVR exposure, or just by type of occupation?

How will personal, non-work time exposure be taken into consideration in the study of cSCC and occupational solar UVR exposure? In order to assess causation in the workplace. I would expect to see how this will be considered in the protocol given the ubiquitous nature of sun exposure.

Has another review similar to the one proposed in this protocol be done before? Or similar? Please mention in the introduction.

Why was the review process started a year ago and the protocol is still under review with a journal? Has the review work not begun yet? It should have waited until the protocol was peer reviewed, in addition to registration on PROSPERO.

Articles from all languages will be included in the systematic review but it is not stated whether there are people in the team who speak all languages around the world, or how this process will be managed.

It would be helpful to know the time frame of the proposed work.

Reviewer #2: It is of importance to critically assess the increased risk of developing cutaneous SCC in outdoor workers due to exposure to solar UV radiation. Such an investigation is difficult to undertake due to the many variables involved but is worthwhile not only in terms of personal health but also for drawing up relevant legislation in the future for worker protection. The proposed systematic review and meta-analysis described in this article will, hopefully, provide useful information to add to that available in various recent published reviews and papers.

Below are various comments for consideration by the authors.

!. Search terms used to identify relevant articles in the academic databases are not given.

2. The Methods were brief. More detail is required as follows.

First the phototype of the potential participants is not mentioned which is of importance as those with darkly pigmented skin would be much less likely than those with fair skin to develop SCC through occupational solar UV radiation exposure.

Secondly the exposure was defined as 1 SED/day. Is this every day throughout the year or only in the summer months where relevant? In addition the dose of UV radiation can be assessed in terms of personal dosimeters, ambient radiation, UV Index or questionnaires relating to work practices/past sunburns. Would all these methods be acceptable?

Thirdly the Outcome is stated as being “medical diagnosis” but this can vary from examination by pathology of a biopsy to that suspected by a General Practioner. Would all these be acceptable?

3. There are a number of small errors throughout such as the superscript 9 in the title which does not relate to an author, 1.8 M cases of cSCC out of 7.7 M MNSCs is the same as 20% so this does not need to be stated twice (lines 6-8, Introduction), the sentence beginning “About the number…..” requires rewriting (beginning of Second paragraph, Introduction), and there are lot of unnecessary hyphens throughout.

7. PLOS authors have the option to publish the peer review history of their article (what does this mean?). If published, this will include your full peer review and any attached files.

Reviewer #1: No

Reviewer #2: No

---

## [Author Response · Author response to Decision Letter 0]

28 Jan 2023

Please refer to the "Response to Reviewers"-Letter. We opted for answering in very detail in a very traceable way there.

---

## [Decision Letter · Decision Letter 1]

7 Feb 2023

PONE-D-22-18577R1Risk of cutaneous squamous cell carcinoma due to occupational exposure to solar ultraviolet radiation: protocol for a systematic review and meta-analysisPLOS ONE

Dear Dr. Wittlich,

Thank you for submitting your manuscript to PLOS ONE. After careful consideration, we feel that it has merit but does not fully meet PLOS ONE’s publication criteria as it currently stands. Therefore, we invite you to submit a revised version of the manuscript that addresses the points raised during the review process.

The replies to the reviewers comments’ have been judged satisfactorily but there are some minor issues that need additional clarifications. I agree with the points raised up by Reviewer #2. Before the submission, please remember to hide the comments you put in Microsoft Word, now appearing in the right column of the track-changed manuscript version.

We look forward to receiving your revised manuscript.

Kind regards,

Matteo Rota, Ph.D.

Academic Editor

PLOS ONE

Journal Requirements:

Reviewers' comments:

Reviewer's Responses to Questions

**Comments to the Author**

1. Does the manuscript provide a valid rationale for the proposed study, with clearly identified and justified research questions?

Reviewer #1: Yes

Reviewer #2: Yes

2. Is the protocol technically sound and planned in a manner that will lead to a meaningful outcome and allow testing the stated hypotheses?

Reviewer #1: Yes

Reviewer #2: Yes

3. Is the methodology feasible and described in sufficient detail to allow the work to be replicable?

Reviewer #1: Yes

Reviewer #2: No

4. Have the authors described where all data underlying the findings will be made available when the study is complete?

Reviewer #1: Yes

Reviewer #2: Yes

5. Is the manuscript presented in an intelligible fashion and written in standard English?

Reviewer #1: Yes

Reviewer #2: Yes

6. Review Comments to the Author

You may also provide optional suggestions and comments to authors that they might find helpful in planning their study.

Reviewer #1: Thank you for addressing my comments. I am happy with the revised manuscript. I have no further comments.

Reviewer #2: Although the revised manuscript is improved compared with the original submission, there remain a number of aspects for consideration by the authors, as follows:

Line 36 and elsewhere – why are there comments in the right hand margin?

Line 38. How many readers of this article will know the current occupational exposure limit for artificial UVR? The actual limit should be stated especially as artificial UVR is not mentioned in the text of the article.

Line 39-40. It is unclear what “continuous and cumulative patterns of sun exposure” means. Suggest replacing this by “chronic and intermittent sun exposure”.

Line 42. Add age and phototype to this list.

Line 50. What does “potentially expected globally” mean?

Line 153. The statement that that “The exposure studied is defined as 1 SED per day which can be achieved by spending more than one hour of the working day by outdoor workers during daylight hours….” cannot be valid for those working outdoors in the winter months, say at latitudes above 45 degrees. See, for example, Soueid et al Br J Dermatol showing that the median daily measured personal UVR dose for an outdoor worker is 1.3 SED in winter at a latitude of 41 degrees north, and would obviously be less at higher latitudes.

Line 156. Is latitude of residence or ambient UVR levels excluded?

Line 161. It is not normal medical practice, at least in the UK, to biopsy Bowen’s or AK lesions. Relying on biopsy results for such lesions would therefore not give a valid picture of their frequency in outdoor workers.

Line 176. Why are Bowen’s and AK not included as search terms?

Line 232. Add body site of lesions (whether on a frequently UV-exposed site or not).

7. PLOS authors have the option to publish the peer review history of their article (what does this mean?). If published, this will include your full peer review and any attached files.

Reviewer #1: No

Reviewer #2: No

---

## [Author Response · Author response to Decision Letter 1]

15 Feb 2023

We adressed all the points in a letter, which has been submitted along with the manuscript.

---

## [Decision Letter · Decision Letter 2]

20 Feb 2023

Risk of cutaneous squamous cell carcinoma due to occupational exposure to solar ultraviolet radiation: protocol for a systematic review and meta-analysis

PONE-D-22-18577R2

Dear Dr. Wittlich,

We’re pleased to inform you that your manuscript has been judged scientifically suitable for publication and will be formally accepted for publication once it meets all outstanding technical requirements.

Kind regards,

Matteo Rota, Ph.D.

Academic Editor

PLOS ONE

Additional Editor Comments (optional):

Reviewers' comments:

Reviewer's Responses to Questions

**Comments to the Author**

1. Does the manuscript provide a valid rationale for the proposed study, with clearly identified and justified research questions?

Reviewer #2: Yes

2. Is the protocol technically sound and planned in a manner that will lead to a meaningful outcome and allow testing the stated hypotheses?

Reviewer #2: Yes

3. Is the methodology feasible and described in sufficient detail to allow the work to be replicable?

Reviewer #2: Yes

4. Have the authors described where all data underlying the findings will be made available when the study is complete?

Reviewer #2: Yes

5. Is the manuscript presented in an intelligible fashion and written in standard English?

Reviewer #2: Yes

6. Review Comments to the Author

You may also provide optional suggestions and comments to authors that they might find helpful in planning their study.

Reviewer #2: The authors have addressed the comments of reviewer 2 to a satisfactory degree. Acceptance of the revised manuscript is recommended.

7. PLOS authors have the option to publish the peer review history of their article (what does this mean?). If published, this will include your full peer review and any attached files.

Reviewer #2: No

---

## [Editor Report · Acceptance letter]

22 Feb 2023

PONE-D-22-18577R2 

Risk of cutaneous squamous cell carcinoma due to occupational exposure to solar ultraviolet radiation: protocol for a systematic review and meta-analysis 

Dear Dr. Wittlich:

I'm pleased to inform you that your manuscript has been deemed suitable for publication in PLOS ONE. Congratulations! Your manuscript is now with our production department. 

Kind regards, 

on behalf of

Prof. Matteo Rota 

Academic Editor

PLOS ONE